# Single-sided magnetic resonance-based sensor for point-of-care evaluation of muscle

Sydney E. Sherman [1,2], Alexa S. Zammit[2,3], Won-Seok Heo [2],
Matthew S. Rosen [4,5,6] & Michael J. Cima [2,3] ✉

Magnetic resonance imaging is a widespread clinical tool for the detection of soft tissue morphology and pathology. However, the clinical deployment of magnetic resonance imaging scanners is ultimately limited by size, cost, and space constraints. Here, we discuss the design and performance of a low-field single-sided magnetic resonance sensor intended for point-of-care evaluation of skeletal muscle in vivo. The 11 kg sensor has a penetration depth of >8 mm, which allows for an accurate analysis of muscle tissue and can avoid signal from more proximal layers, including subcutaneous adipose tissue. Low operational power and shielding requirements are achieved through the design of a permanent magnet array and surface transceiver coil. The sensor can acquire high signal-to-noise measurements in minutes, making it practical as a point-of-care tool for many quantitative diagnostic measurements, including T2 relaxometry. In this work, we present the in vitro and human in vivo performance of the device for muscle tissue evaluation.

Point-of-care (POC) medical diagnostics are increasingly utilized in both inpatient and outpatient settings[1,2]. The ability to rapidly detect aneurysms, fluid pockets, and other clinical findings that can be managed using an interventional procedure can decrease the time to diagnosis and treatment, leading to improved patient outcomes[3,4]. The bedside operation of these POC instruments enables measurement of diagnostic information without the need to transport the patient to a centralized-care facility – reducing cost, time to treat, and in some cases, length of stay[5].

Magnetic resonance imaging (MRI) is the primary clinical tool for detecting soft tissue pathology due to high soft tissue contrast. It is non-invasive, does not involve patient exposure to ionizing radiation, and allows for quantification of tissue morphology. Traditionally, MRI is not practical as a POC tool since the high magnetic fields (typically 1.5–3 Tesla) needed for operation present a projectile hazard for ferrous objects if operated outside of an access-controlled scanner suite. Additionally, the need for magnetic and radio frequency (RF) shielding, as well as power requirements that can exceed 25 kW, increase the

footprint precluding use at the POC. It is also not compatible with patients that have certain types of metal implants; the high cost of scanner purchase and site infrastructure limitations prevent many facilities from having multiple scanners, limiting capacity despite high demand for instrument use.

Recent technical innovations in MRI physics and instrumentation have led to scanners operating at far lower magnetic fields than previously thought possible and have enabled 64 mT MRI scanners to be deployed at the patient bedside for POC use. These low-cost low-field MRI scanners can operate without the shielding and safety requirements of traditional high-field scanners, but their use to date has focused on neuroimaging in critical-care settings[6–9].

Single-sided magnetic resonance (SSMR) sensors may provide a portable POC diagnostic option that leverages the power of MR-based contrast with purpose-built low-cost hand-held instruments[10]. These devices use magnetic resonance techniques to acquire spectroscopic (i.e., non-imaging) data over a limited tissue depth but have the ability to distinguish between tissue types, intra- and extra-cellular

[1]Harvard-MIT Program in Health Science and Technology, Massachusetts Institute of Technology, Cambridge, MA 02139, USA. [2]Koch Institute for Integrative Cancer Research, Massachusetts Institute of Technology, Cambridge, MA 02139, USA. [3]Department of Materials Science and Engineering, Massachusetts Institute of Technology, Cambridge, MA 02139, USA. [4]Department of Radiology, Athinoula A. Martinos Center for Biomedical Imaging, Massachusetts General Hospital, Boston, MA 02129, USA. [5]Harvard Medical School, Boston, MA 02115, USA. [6]Department of Physics, Harvard University, Cambridge, MA 02138, USA. ✉e-mail: mjcima@mit.edu

compartments and provide information about tissue architecture[11]. Analysis spatially resolved T2-relaxation data from MRI has shown that the skeletal muscle compartment and the subcutaneous compartment can be represented by biexponential decays[10]. There is extensive evidence of skeletal muscle quantitative T2 relaxation being better represented by a bi-exponential model as compared to a mono-exponential model. There is open discussion as to the specific physiological context of the two exponential decays, with support for the two relaxation dynamics originating from either water and lipids, or from differing water compartments within a tissue[12–14]. Multi-compartment analysis has demonstrated higher specificity in differentiating muscle tissues with inflammatory pathologies, dystrophic pathologies, differing fat fractions, and differing water content, regardless of the origin of bi-exponential signal[12,13]. We support the conclusions of previous work that the two relaxations represent intercellular (shorter component) and intracellular (longer component) water compartments within a singular tissue. While both muscle and subcutaneous tissues exhibit biexponential T2 decays, the shorter 'intracellular' time constant is largely conserved between tissue types, while the longer of the time constants differs between tissues and can be distinguished from one another[14].

Techniques including T2 relaxometry and T2-weighted diffusion can be performed on single-sided sensors to provide clinically-actionable information[15–18]. Uses include assessment of liver disease, inflammation, tumor characteristics, iron overload, and cartilage diseases, among others[6,11,19]. Within skeletal muscle tissue specifically, relaxometry can provide insight into fluid status, progressive disease musculoskeletal disease monitoring (sarcopenia, muscular dystrophies, etc.), vascular kinetics and oxygenation tracking, among other applications[10,13,20]. Portable MR sensors are often constructed from permanent magnets to reduce the power requirements. Further, single-sided sensor designs do not require patient movement between rooms or beds to fit inside a magnet bore[17,21–29]. Previous clinical studies with SSMR sensors were limited by several factors, including penetration depth (<6 mm), signal sensitivity, and engineering challenges that would affect subject safety[10]. This prevented human in vivo demonstration of the diagnostic utility of SSMR devices that was demonstrated in prior ex vivo and murine studies. No one clinical application is evaluated for the use of this tool. Outcome metrics, acquisition parameters, and analysis techniques for specific applications will vary based on clinical application.

We demonstrate here the design and utility of a low-field single-sided MR sensor intended for POC evaluation of skeletal muscle in vivo[20,30]. The penetration depth is larger than other single-sided systems permitting accurate analysis of muscle tissue and avoiding signal contribution from other subcutaneous layers, including adipose tissue. Low operational power and minimal shielding requirements are achieved by constructing a permanent magnet array and surface RF coil. The sensor can acquire high SNR measurements in minutes, making it practical as a POC tool. We characterize the in vitro performance of the instrument using several multi-layer tissue phantoms. Finally, we demonstrate the sensitivity of the device to muscle sensing in vivo on a cohort of healthy human subjects.

## Results

### Magnet array design and construction

The ability to unambiguously measure magnetic relaxation properties in muscle tissue without interference from more proximal features would aid clinical evaluation of fluid volume status and other applications. There are experimental challenges to realizing such a system. The homogeneous magnetic field region of a single-sided MR system defines the sensitive region (sweet spot) of the sensor. This must be far enough above the surface of the sensor to predominantly lie within muscle tissue for practical in-vivo use. The thickness of subcutaneous adipose tissue layers varies with body location, so the operation of a

compact single-sided NMR sensor may be limited to those anatomical locations where the sensor's "sweet spot" depth can reach the muscle. There are select locations where, even on subjects with a higher body mass index (BMI), the subcutaneous layer rarely exceeds 6 mm[10,31,32]. We designed our sensor to operate in these types of anatomical locations, specifically the gastrocnemius (calf) muscle.

We performed a finite-element analysis using COMSOL Multiphysics software (Burlington, MA) to design the attainable magnetic fields of the permanent magnet sensor. The net magnetic field profiles of several magnet array configurations were simulated to achieve a design that met our requirement for a POC sensor. Specific endpoints, in order of priority, included homogeneous region depth from the magnet surface (>8 mm), homogeneous region strength (>0.2 T), homogeneous region volume (>200 mm$^3$), weight (<12 kg), and final magnet array size. A two-step computational approach guided the design of individual magnet orientations and configurations of the individual magnets as well as the presence, size, and positioning of iron yokes to shim the magnetic flux. We utilized the unilateral linear Halbach array and cylindrical Halbach array configurations as the basis for our design. Individual feature parameters were swept to achieve a large volume homogeneous region while minimizing stray fields outside of that region. The position of the iron yoke on the inner sides of the raised Halbach elements had the strongest effect on increasing the area, depth, and strength of the field. The raised Halbach elements on the ends of the array and the magnetic orientation, particularly of the outer slices of magnets, similarly had a strong effect on the final profile. The magnet array of the sensor device (Fig. 1) is designed to comfortably seat the calf muscle, allowing for muscle measurements with less variability due to leg placement against the sensor.

The magnet array is constructed from half-inch cube N52 neodymium magnets deployed in machined aluminum frames. The magnets used in the final array were pre-screened and selected for homogeneity. Iron yokes with thickness of 1-mm were placed on the inner raised Halbach elements to increase the volume of the homogeneous region. The mapped homogeneous region following fabrication has a maximum field strength of 0.2 T (8.516 MHz) at the surface of the sensor with a natural descending gradient in the Z direction of approximately 1 T/m and a homogeneous region that sits 8 mm above the surface of the array (field strength 0.196 T). A single surface transceiver coil is used for the pulsed magnetic field[33,34]. A surface coil placed directly in contact with the magnet array, seated on a 0.5 mm layer of Aluminum Nitride, allows the calf of a subject to be placed directly in contact with the sensor and is agnostic to body size and placement. The assembled sensor with Delrin casing weighs 11 kg and is 22 cm length × 17.4 cm in width, and 11 cm in height.

### Clinical design aspects

The ability to acquire data with high SNR enables short acquisition times which is an important clinical consideration for SSMR sensors. The sensors must maintain high signal fidelity in different environments such as hospitals, outpatient centers, homes, etc. in the presence of a variety of other equipment generating uncontrolled EMI interference.

Inductive and capacitive coupling between the RF coil and the subject can increase the RF noise, making reliable grounding strategies critical. The Delrin casing around the completed magnet array was designed with six copper grounding rods with 5-mm-diameter that contact the bottom surface of the aluminum frame of the magnets on one end and contact with the external aluminum plate on the other end, where the sensor is placed. The copper rods in our design sit on a grounded aluminum plate to ensure the magnets are grounded. The 'L' impedance matching network is set in a 3D printed polylactic acid (PLA) casing with a thin aluminum housing. This allows a human subject to be grounded through contact of the leg with a metal casing around the network. The grounding configuration significantly

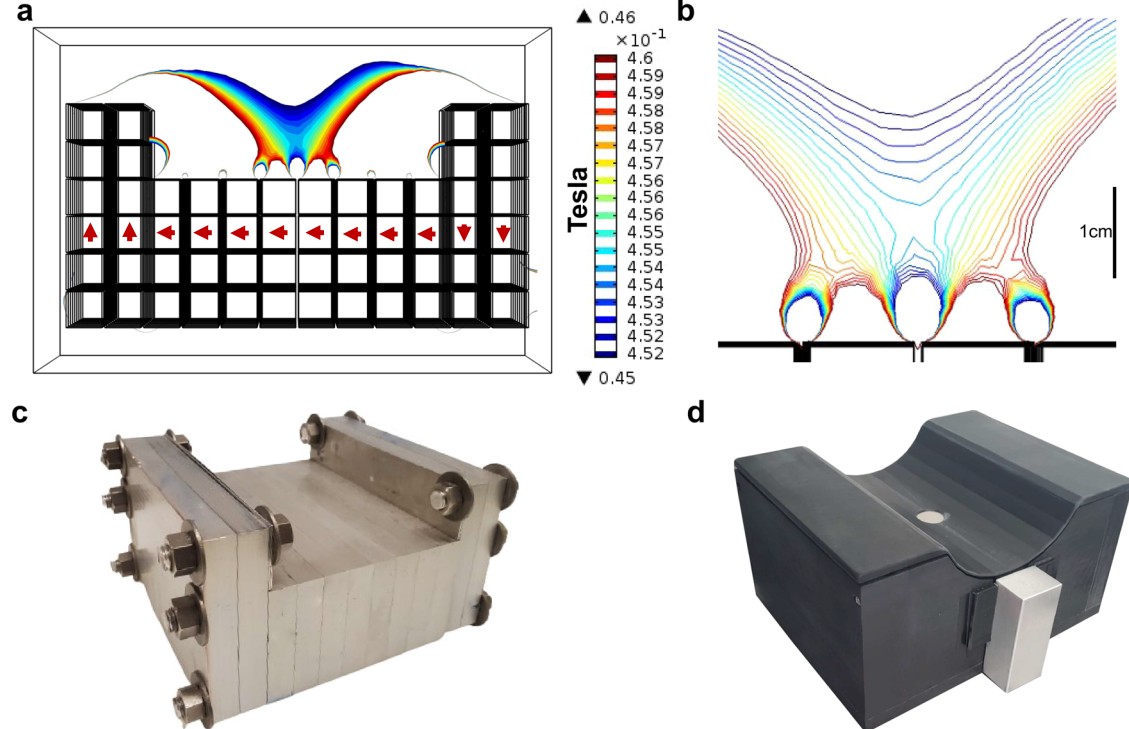

**Fig. 1 | Magnet array design and construction. a** Simulated magnetic field profile of magnet array, each box is one 12.7 mm cube magnet, red arrows indicate magnet orientation **b** Simulated magnetic field profile of homogeneous region with scale bar **c** Constructed magnet array with assembled aluminum frames and iron yokes. **d** Assembled magnet array with RF coil, matching network, and Delrin casing. Source data are provided in a Source data file.

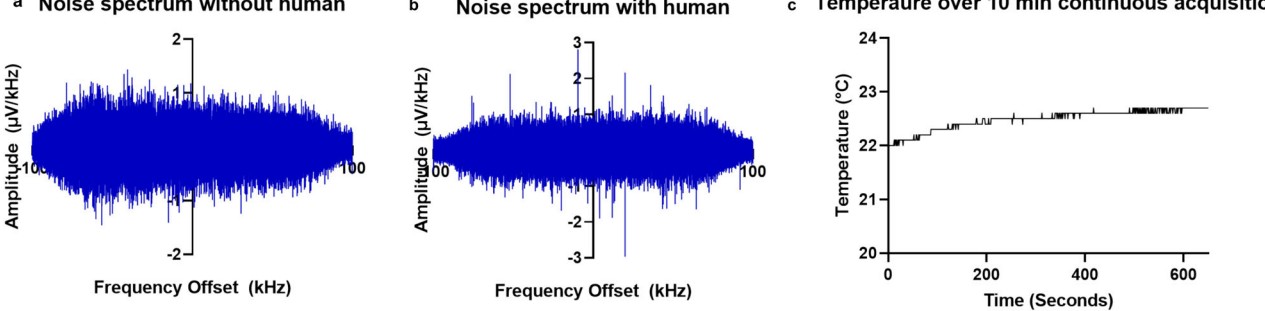

**Fig. 2 | Sensor noise and temperature performance in clinical environment. a** Noise spectrum from transceiver coil without human in contact with the sensor, root mean square (RMS) noise = 0.22 μv. **b** Noise spectrum from transceiver coil with human in contact with sensor, RMS noise = 0.25 μv. **c** Temperature increase above coil encased by aluminum nitride over a continuous 10-minute acquisition. Source data are provided in a Source data file.

decreases baseline noise for human subjects. Noise levels are approximately 0.22 μV without a human subject and 0.25 μV with a human subject in contact with the sensor (Fig. 2a, b).

Transceiver coil heating is a critical concern for the clinical implementation of SSMR[35,36]. High-power RF pulse trains generate Joule heating which causes the coil to heat to a temperature that must be mitigated for human use. The acquired signal is insufficient for meaningful measurements within reasonable scan times if the pulse power is decreased. This is addressed by conducting the heat to the magnet through a thermally conductive path. Aluminum nitride (AlN) is a high thermal conductivity dielectric. AlN sheets (0.25–1.2 mm thick) were machined to fit around the coil, with an additional 0.25 mm sheet placed over the top of the coil. The AlN sheets direct the heat generated by the pulse trains away from the human subject and into the body of the aluminum magnet frame. The AlN sheets surrounding the coil also prevent human subjects from coming in direct contact with the coil. This approach prevents sample heating of more than 1 °C over 10 min of continuous signal acquisition, as displayed in Fig. 2c.

## In vitro sensor characterization

A CPMG pulse sequence was used to acquire T2 relaxometry data. Characterization of the sensor's depth sensitivity was achieved with copper sulfate phantoms. Slice selection characterization was performed by adjusting the pulsed (B1) frequency from 8.32–8.42 MHz in 0.01 MHz increments. A 1 M copper sulfate sample was measured at each frequency in a machined PEEK case with a 1 mm sample height. The sample was raised in 1 mm increments perpendicularly above the surface of the array and signal was acquired at each sample position. The relation between B1 frequency and depth sensitivity is shown in Fig. 3.

Custom tissue phantoms were made to more accurately evaluate the ability of the sensor to capture signals from muscle tissue while avoiding signals from subcutaneous adipose tissue. These phantoms consist of two compartments, oil, and water, emulsified with different percentages of the components, the size of the phantom (27 mm diameter) is larger than the size of the coil to mimic calf geometry exceeding the size of the coil[37]. The phantoms mimic the intra- and

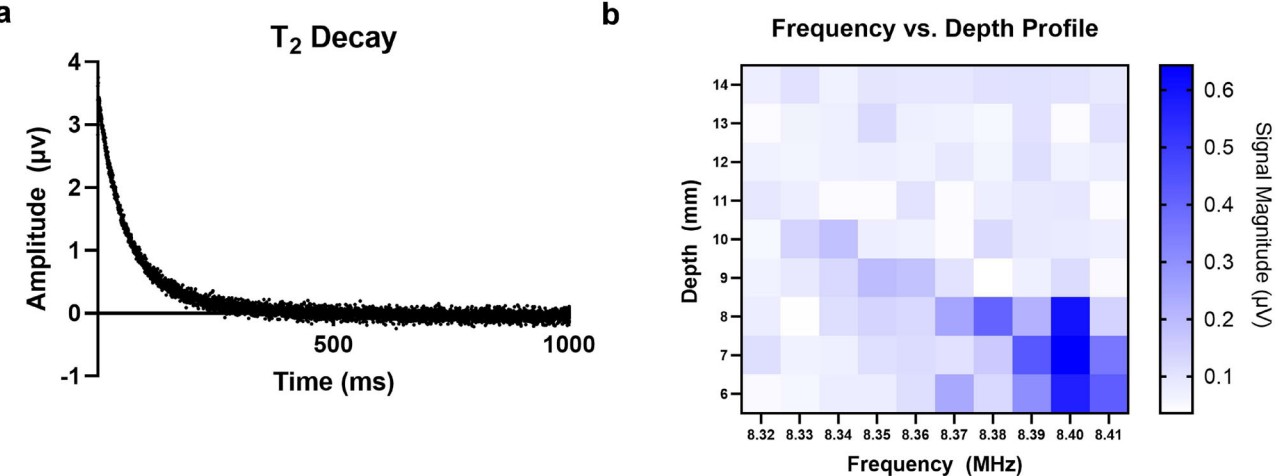

**Fig. 3 | Field characterization. a** Representative T2 decay from sensor. **b** Depth sensitivity profile as a function of B1 frequency using a 1 mm slice copper sulfate phantom. Source data are provided in a Source data file.

inter-cellular compartments of both muscle and adipose tissue. They have similar T2 relaxation times and differing relative amplitudes when analyzed with a bi-exponential fit due to the different contributions of the oil and water. Individual phantoms were created for adipose tissue only, muscle tissue only, and layered adipose (6 mm thick) with muscle tissue over. Signals from the adipose and muscle tissue phantoms were used as the standard for the subsequent layered phantom tests. The adipose layer of the phantoms can be poured with variable thicknesses to mimic differing amounts of subcutaneous adipose tissue between the surface of the magnet array and the skeletal muscle from which we obtain signal measurements.

Data was collected from the phantoms at 8.48, 8.43, 8.38, and 8.29 MHz, corresponding to depths of approximately 2-, 5-, 8-, and 10-mm from the surface of the sensor. The relative signal magnitude of the second component of a biexponential fit of the T2 decays verify, shown in Fig. 4, verifies the slice selectivity and ability to capture the signal from deeper skeletal muscle phantom while avoiding the 6-mm-thick subcutaneous adipose tissue layer, shown in Fig. 4. We select the sampling region of the signal based on the permanent magnetic field profile by altering the frequency of the RF pulses. Decreasing the frequency allows for signal sampling further from the surface of the magnet array and deeper into the tissue. The signal from the layered phantom acquired at and below 8.38 MHz is statistically the same as the muscle tissue phantom. This demonstrates we are only capturing the signal from the muscle portion of the layered phantom. The signal reflects amplitudes between the muscle and adipose phantoms at 8.43 MHz. The signal at this frequency contains contributions from both phantom types near the layer junction. The signal acquired at 8.48 MHz, however, statistically reflects the adipose phantom; verifying that we are fully below the phantom layer junction. We can achieve an accurate signal from muscle phantom above a 6-mm-thick layer of adipose phantom at 8.38 MHz using a biexponential fit of the decay.

### In-vivo sensor performance

Subjects placed their leg on the sensor and were scanned for 10 minutes to produce a high SNR ( > 150) signal. Separately, adipose (axillary and inguinal site) and muscle (gastrocnemius and soleus) tissue were excised from rodents by a veterinary technician. This ex-vivo murine tissue was placed directly on an SSMR sensor to obtain 'standard' SSMR relaxometry measurements for muscle tissue and adipose tissue individually. In-vivo human measurements were compared to the ex-vivo murine tissue to determine the ability of the device to capture in-vivo signals from muscle tissue. The data was analyzed with bi-

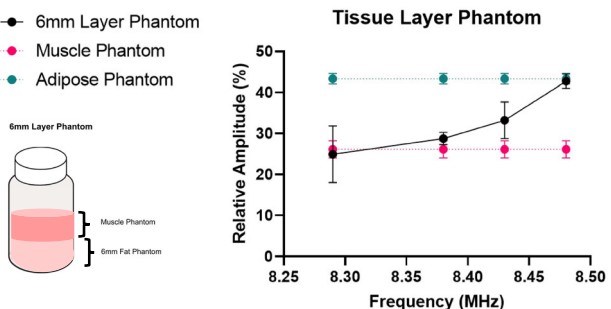

**Fig. 4 | In-vitro layered phantom characterization.** Biexponential fit T2 values and amplitudes of muscle phantom, adipose phantom, and 6 mm adipose layer phantom acquired at 8.29, 8.38, 8.43, and 8.48 MHz. Three measurements from the same phantom were acquired in random order. Data is presented as mean +/- SD. Source data are provided in a Source data file.

exponential fit to determine if the human data captured resembles muscle tissue. The biexponential fit captures the intra- and extra-cellular compartments of tissue. There is a statistically significant difference between the in-vivo human signal and the ex-vivo adipose signal for both the T2 times and relative amplitudes. Three of the four values comparing in-vivo human signal and ex-vivo muscle tissue have statistical similarity, demonstrating that we successfully capture signals from in-vivo muscle tissue of human subjects (Fig. 5).

### Discussion

The ideal diagnostic tool provides rapid and actionable information to clinicians without disruption to the patient. MRI is a common non-invasive diagnostic tool for soft tissue, but except for recent low-field instantiations, the size, cost, and safety considerations of these machines significantly limit POC use. Single-sided MR tools can provide localized NMR measurements close to the surface of the sensor. Depending on the RF pulse sequence utilized, these sensors can be sensitive to T1 or T2 relaxation and diffusion. These methods provide different information about tissue architecture and pathology.

Single-sided MR sensors are not currently used in clinical practice. Several limitations exist, including the measurement depth, acquisition time, and validation of signal sensitivity for clinical decision making. To date, no single-sided sensor has been designed with the penetration depth and low gradient necessary for clinical implementation, especially for single-voxel measurements. We maximize

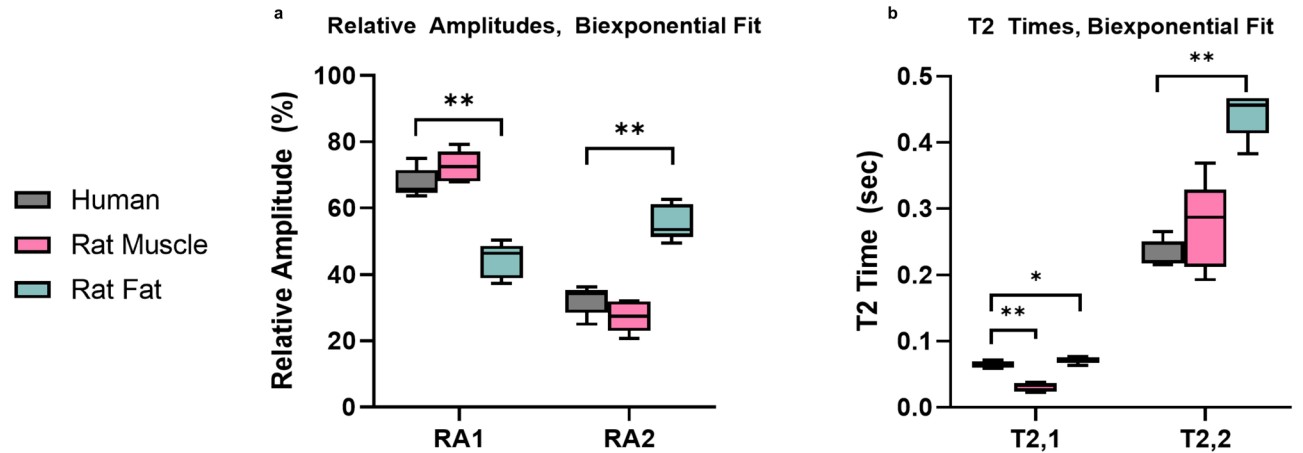

**Fig. 5 | Comparison of in-vivo human subjects and excised murine muscle and fat tissue. a** Biexponential fit, relative amplitudes (**$p = 0.000035$). **b** Biexponential fit T2 times. (*$p = 0.0476$, **$p = 0.00000921$, **$p = 0.00000138$) Five measurements were taken from distinct animal samples or human subjects. Box bounds indicate upper and lower quartiles, center line indicates median, whiskers indicate data minima and maxima. Single tailed $t$-tests were used to determine significance. Source data are provided in a Source data file.

the signal acquisition region by minimizing the gradient. This is accomplished by raising the edge Halbach elements of the magnet array into more of a semi-cylindrical geometry and positioning iron yokes on the inner surfaces of these edge elements, which collectively reduce the gradient in the homogeneous region while maintaining its depth from the array surface. Other single-sided sensors have significantly higher magnetic field gradients, which increases slice selectivity but reduces the signal intensity and sensitive volume.

We demonstrate the design and construction of an MR sensor for in vivo detection and evaluation of skeletal muscle. Our single-sided MR sensor was designed with permanent magnets to have an easy to fabricate static $B_0$ magnetic field. The magnet array is a unilateral Linear Halbach design with raised edges which offers a remote, low field, low gradient, homogenous region. The Halbach array is well suited for POC settings because the magnetic field is limited to only one side of the array, eliminating the effects of stray fields and the need for specialized shielding and shimming. The sensitive region of this sweet spot magnet is 8 mm from the surface of the magnet. The magnetic field produced by this design is aligned parallel to the surface of the sensor. That field direction permits a standard surface transceiver coil on the surface of the magnet. This region contains muscle when placed against a human's leg. The magnet array produces a magnetic field gradient that decreases in the direction of depth into the tissue. The frequency of the RF pulse required to excite a proton depends on $B_0$ field strength. A Car-Purcell-Meiboom-Gill (CPMG) Pulse sequence is used for signal acquisition to maximize the SNR due to its insensitivity to $B_0$ field homogeneities. The sensor was designed to acquire high SNR signals with high-power RF pulses without inducing discomfort caused by coil heating through the inclusion of aluminum nitride sheets encasing the RF coil. Grounding lines for the portable system are constructed within the array casing to reduce external wiring, provide reproducible grounding solutions, and increase flexibility and portability.

The CPMG sequence captures T2 relaxation times and amplitudes, which allows for tissue analysis. T2 times are tissue compartment dependent as this relaxation property is the function of the microstructure and composition of the tissue. Proton spins are the origin of the T2 signal, almost all of which are from water. Different characteristics of water within the body result in different T2 times; the protons in pure water, water bound to macromolecules, interstitial, and intracellular water all diphase at different rates which allows them to be separated and quantified. This measurement was investigated previously for in-vivo fluid estimation, and tissue identification, and can be used for several other tissue architecture questions.

We created tissue phantoms to validate the sensitivity of our sensor. Layers of adipose and muscle in a multi-phase phantom were fabricated to demonstrate the acquisition of the muscle "signature" without collecting signals from the more proximal adipose layer. We then demonstrated our device's ability to collect a muscle signature from a cohort of healthy volunteers. Subjects placed their calf on the sensor from a seated position. A 10-minute acquisition time was sufficient to acquire a high SNR signal (>150) in muscle tissue. The T2 times and relative amplitudes obtained from the fit reflect expected values for muscle tissue in direct comparison to excised murine muscle and adipose tissue. There are potential usage limitations for muscle detection in subjects with body composition consisting of thicker subcutaneous adipose layers over the gastrocnemius muscle. Additionally, due to B1 inhomogeneity and increased strength affecting more proximal tissue than the sweet spot, it is like to assume some signal influence from these features. Ultimately, our intention is to minimize scan time while also reliably producing an SNR of 200. This can be acquired over approximately 6 min based on our initial data and we anticipate further SNR increases which would continue decreasing the necessary acquisition time.

This paper presents the design and construction of an SSMR sensor for in vivo muscle detection. The permanent magnet array creates a 0.2 T homogeneous region lifted 8 mm above the surface of the magnet. A small surface coil transmits high-power pulses in long echo trains enabling T2 relaxometry measurements. Tissue heating is avoided by encasing the coil in aluminum nitride. Our sensor was characterized and validated with complex layered tissue phantoms representing subcutaneous adipose tissue and muscle. Finally, we tested the sensor on a small cohort of healthy human subjects to determine if we could successfully detect muscle tissue. Future studies include additional evaluation of diagnostic utility and design and application of complex RF pulses.

## Methods
This research complies with all relevant ethical regulations. All animal studies were overseen by the Massachusetts Institute of Technology Committee on Animal Care, protocol 2208000409. Human subject studies were overseen by the Massachusetts Institute of Technology Committee on the use of Humans as Experimental Subjects (COUHES), protocol 2002000099.

## FEA design

The desired single-sided magnet array has a sweet spot (homogeneous region) field strength of at least 0.2 T lifted at least 6 mm above the surface of the magnet. COMSOL Multiphysics was used to perform finite element analysis. The permanent magnetic field was generated using the magnetic fields, no currents module which uses $-\nabla \cdot (\mu_0 \nabla V_m - \mu_0 \nabla M_0) = 0$ to generate the field profile (Fig. S1). Remanent flux density and relative permeability were determined by the values for N52 neodymium magnets (1.48 T and unity). An extra-fine physics defined mesh was applied to the geometry to solve Gauss' Law. The magnet array geometry was placed inside a large rectangular prism with no magnetization and air to provide the simulation environment. Sweet spot volume ($\epsilon = 0.5\%$), strength (T), and depth (mm) were used as outcome measures.

A two-step approach was used to determine array configuration. Initially 2-D models were used to approximate the net magnetic profile of several basic magnet array geometries including unilateral Halbach, semi-cylindrical Halbach, U shaped, and L shaped For these geometries, an optimization score was calculated based on the B0 strength and gradient: $Optimization\ Score = \frac{B_0^{\frac{7}{4}}}{RMSE \cdot G_0}$[35]. The Optimization score and depth of homogeneous region was used to select a semi-cylindrical design as the basis for further design optimization. A 3-D model of the geometry was then used to evaluate several parameters including magnet size, shape, orientation, spacing, inset depths, and iron yoke position and determine the effects of the parameters on the magnetic field profile. The position of the iron yoke on the inner sides of the raised Halbach elements had the strongest effect on increasing the area, depth, and strength of the field. The raised Halbach elements on the ends of the array and the magnetic orientation, particularly of the outer slices of magnets, similarly had a strong effect on the final profile. Features including the size of the magnet cubes, central slice, and inset depths of the central slice had opposing effects on the net field strength, area, and depth.

## Fabrication

The final magnet array design consists of 448 individual N52 (Nd1Fe14B) magnets. Each magnet is 12.7 mm cubed. One thousand serialized magnets (Viona Magnetics, Hicksville, NY) were individually flux tested with a hall probe and gauss meter (Lake Shore Cryotronics, Woburn, MA). Of those, the most homogeneous were selected for inclusion in the constructed array.

Each magnet was secured in a machined aluminum frame (Xometry, North Bethesda, MD). Two frame styles were designed, the center style holds 32 magnets arranged in 4 rows of 8, and the end style holds 48 magnets in 6 rows of 8. In total, 4 end and 8 center frames were machined. Cube magnets were placed in the aluminum frame and covered with a temporary aluminum cover slip. The cover slip prevented individual magnets from ejecting, due to repulsion, from the frame during subsequent magnet placement. All magnets in a frame are oriented in the same direction according to the Fig. 1. There is a 1 mm gap between each of the magnets in the frame.

A temporary structure was made using three-foot aluminum rods secured in a 6 in × 12 in aluminum block to align the magnet frames together. The bolt holes on each array frame were used to slide the frames along the rods without allowing for sideslipping in any direction due to the strong magnetic forces. The temporary cover slips over the magnets were removed once all 12 frames were aligned on the rods. The aluminum rods were removed one at a time and replaced with brass bolts. Machined Delrin (Xometry, North Bethesda, MD) is used to encase the aluminum magnet array and block stray magnetic fields. The fabrication process is shown in Figure S2.

A transceiver coil was constructed from AWG32 magnet wire (MWS Wire, Oxnard, CA) wound around a cylindrical Teflon former (McMaster Carr, Elmhurst, IL, #91182A170). The coil has 8 turns and is 16 mm in diameter. The coil is connected to an 'L' impedance matching network with capacitances selected for our desired frequency range (8.2–8.5 MHz).

## Phantom construction

Multi-phase tissue phantoms were fabricated according to the general protocols described in Bush et al.[37] This emulsion phantom protocol allows for multi-component analysis and more accurate physiological features. The protocol was modified to fabricate phantoms for different tissue types by adjusting the percentage of oil and aqueous components. A 40% oil fraction was used for muscle tissue and a 70% oil fraction was used for adipose tissue.

The aqueous phase components consist of deionized (DI) water, sodium benzoate (Sigma-Aldrich, St. Louis, MO, 71300), Tween-20 (Sigma-Aldrich, St. Louis, MO, P1397), and agar (Sigma-Aldrich, St. Louis, MO, A1296). To prepare 100 mL of the aqueous phase, 100 mL of DI water was added to a 400 mL beaker. The beaker was placed on a hotplate set at 90 °C with a stir rate of 100 rpm. 0.1 g of sodium benzoate was measured and added to the water, followed by 0.2 mL of the water-soluble surfactant. Next, 3.0 g of agar was slowly added to the water beaker. Once added, the hotplate temperature was increased to 350 °C and the stir bar speed increased to 1100 rpm for 5–10 min to melt the agar. The solution was removed from the hotplate to check for clear color, no dispersed air bubbles, and no clumps or streams of agar. The aqueous solution was tested to ensure the agar melted by placing about 5 mL of solution in a separate glass vial. If the solution set and was clear, then the solution was then placed back on the hotplate (50 °C and 100 rpm) while the oil solution was prepared. If the separated solution did not set, the hotplate temperate was increased and the agar was given more time to melt before proceeding.

The oil solution consists of peanut oil (Sigma-Aldrich, St. Louis, MO, P2144) and Span 80 (Sigma-Aldrich, St. Louis, MO, S6760). To prepare 100 mL of the oil solution, 100 mL of peanut oil was measured and placed in a clean beaker with a clean stir bar. The beaker was placed on a hotplate set at 90 °C with a stir rate of 100 rpm for 1 min. 1.0 mL of the oil-soluble surfactant was added dropwise to the beaker with peanut oil. The hotplate settings were increased to 150 °C and 1100 rpm for 5 minutes to fully mix the oil solution.

To create the phantom emulsion, a clean stir bar was placed in a 250 mL Erlenmeyer flask. A volumetric pipette was used to add the appropriate amount of the aqueous solution to the flask (amount of solution added depends on oil fraction of phantom being created). For example, to create 100 ml of a 40% phantom, 60 ml of aqueous solution was added to the flask. The flask was placed on a hotplate set at 90 °C and 1100 rpm. After 2 min of stirring, 40 ml of the oil solution was measured with a volumetric pipette and slowly added dropwise (around 1 drop per second for emulsions at a fat fraction of 35% or greater) to the aqueous solution in the flask. When streaks of oil were observed in the emulsion, no further oil was added until stirring had fully emulsified the separated oil. Once all the oil solution was added, the hotplate settings were adjusted to 300 °C and 1100 rpm and the emulsion was stirred for 5 min. The emulsion was white, with a creamy and smooth texture with no visible separated oil. The emulsion was then poured into glass vials to cool and set.

## Layered phantoms

Layered phantoms were constructed using the protocol outlined above. To mimic a human leg on a single-sided sensor, the phantom required an adipose layer closer to the surface of the magnet and a muscle layer above. Adipose tissue phantoms were constructed, poured into a vial with depths ranging between 1 and 8 mm, at 1 mm increments, and set overnight in a refrigerator. The following day, muscle phantom was created and poured in a layer immediately on top of the adipose phantom, and re-set in a refrigerator. Melting, phase separation, or mixing of the two phantoms layers was not observed.

Layered phantoms were created for varying levels of subcutaneous adipose layer thickness from 1 to 8 mm.

## Signal acquisitions
A Kea2 spectrometer (Magritek, Wellington, New Zealand) is used to acquire signal. Prospa software V3.100 (Magritek, Wellington, New Zealand) provides the setup and analysis interface. The internal spectrometer RF amplifier is connected via coaxial cable to the matching network.

T2: T2 relaxometry data acquisition was accomplished with a CPMG sequence with 8192 echoes, 12 μs pulse length (43 kHz excitation bandwidth), 65 μs echo time, and 16 acquisition points.

## Mapping
The magnetic field profile of each individual magnet frame, and the final constructed array were characterized with a hall probe (HMMY-6J04-VR, Lake Shore Cryotronics, Woburn, MA) and gaussmeter (Model 475 DSP Gaussmeter, Lake Shore Cryotronics, Woburn, MA). A 16 × 16 × 32 mm area in the center of the array containing the array sweet spot was scanned at 1 mm intervals (Figure S3). The sweet spot ($\epsilon = 0.5\%$) within the mapped region has a volume of 67.5 mm$^3$ with a slice thickness of 1.5 mm in the z plane. The slice thickness is matched with the pulse bandwidth.

## CuSO$_4$ sensitivity profile
Profiles for signal sensitivity vs. depth from the surface of the magnet were performed to determine the optimal frequency for signal acquisition. A PEEK holder was machined with a 1 mm × 16 mm × 32 mm pocket. The pocket was filled with 0.1 M CuSO$_4$ (0.1 M, Sigma-Aldrich, St. Louis, MO, 1.02784), secured to a robotic arm, and positioned directly above the top of a surface RF coil placed on the center of the magnet array.

The sensitivity profile was performed by scanning the CuSO$_4$ (0.1 M, Sigma-Aldrich, St. Louis, MO, 1.02784) sample along a perpendicular line at distances of 6-14 mm above the surface of the magnet in 1 mm increments. This process was repeated using B1 frequencies from 8.32–8.42 MHz in 0.01 MHz intervals.

Each scan was acquired with a Kea2 spectrometer (Magritek, Wellington, New Zealand) using a CPMG pulse sequence with 8192 echoes, 65 μs echo time, 12 μs pulse duration.

## Ex-vivo
All animal studies were overseen by the Massachusetts Institute of Technology Committee on Animal Care. Muscle (gastrocnemius and soleus) and adipose (axillary and inguinal) tissue was excised from rats by a veterinary technician immediately following euthanasia (MIT CAC protocol 2208000409). The tissue samples, obtained from the Koch Institute Swanson Biotechnology Center Animal Imaging and Preclinical Testing Core, were excised from two Sprague Dawley (Male, 5 months) and two WISTAR (Female, 18 months) rats. Excised tissue was wrapped in phosphate buffered saline (PBS) (Sigma-Aldrich, St. Louis, MO, P4474) soaked gauze, placed on ice, and transported for immediate MR characterization. Ex-vivo murine tissue MR characterization was performed on a different SSMR than the one described in this manuscript. The sensor used, described in Colucci et al., functions at a different operating frequency. The same pulse sequence and data fitting methods were used on the two instruments[10].

Signal from 26 different CuSO4 Phantoms at concentrations between 0.001 M and 0.2 M were acquired on the two SSMR sensors to compare the T2 relaxation times between the two sensors. The relation between the T2 times on each magnet array, Fig. S4, was used to perform a direct comparison between ex-vivo data captured on the existing magnet array and the in-vivo data captured on the magnet array described in this manuscript. We expect the shorter T2 times on the existing sensor due to greater influence from the less homogeneous field and T2*. T2 relaxation times of tissue acquired on the sensor described in Colucci et al. were offset using the equation of $y = 1.861 + 0.7377x + 0.006527x^2 (R^2 = 0.9996)$ where $x$ is the relaxation time on the sensor reported by Colucci et al.[10]. And $y$ is the relaxation time on the sensor reported in this study. This allows for a direct comparison to the signal acquired on the sensor described in this manuscript.

## Human experiments
Human subject studies were overseen by the Massachusetts Institute of Technology Committee on the use of Humans as Experimental Subjects (COUHES). Five human subjects were recruited for the study through emails sent to multiple collegiate sports teams. Informed consent was obtained from all human subjects prior to study participation. All consented subjects were female with ages: 18, 23, 23, 25, and 27. Subjects were compensated with $40 amazon giftcards. Sex is not expected to impact results, both sexes were included in recruitment efforts and no sex-based analysis was conducted. Human subjects are asked to sit and place one leg on the MR sensor and the other leg next to the sensor. For each human subject, the matching network is tuned to minimize impedance at our working frequency, and baseline noise with the human subject is recorded. A CPMG pulse sequence, described under Signal Acquisition methods, is used to acquire T2 decay signals. Individual scans of 16 averages take approximately one minute to complete; each individual scan is repeated 10 times, for a total data collection time of nearly 10 minutes for each subject. (MIT IRB protocol 2002000099).

## Fitting
$T_2$ decay curves obtained from each scan are modeled as bi-exponential signals. The $T_2$ decays from each time point were averaged together using a straight-averaging technique. The first three points were deleted from the averaged decay. SNR was calculated as the ratio of the maximum (taken as the average of the first 10 points) value of $T_2$ decay curve divided by the SD of the noise floor at the end of the $T_2$ decay. A nonlinear least-squares fitting method with a trust-region algorithm was used to perform the fits using MATLAB R2022b. A general multi-component exponential decay signal is represented as:

$$y(t, A, \tau) = \sum_{i=1}^{N} A_i e^{-t/\tau_{2,i}}$$

where $y(t)$ is the estimated signal, $N$ is the number of components, $A$ is a vector of amplitudes and $\tau$ is a vector of corresponding relaxation times. Relative amplitudes were obtained from summing the amplitudes for each fit:

$RA_i = \frac{A_i}{A_1 + A_2} \times 100$. The effect of SNR on the bi-exponential fitting results is demonstrated in Supplementary Fig. 5.

## Statistical analysis and reproducibility
Statistical analysis for comparison among groups was performed in MATLAB R2022b. All tests were single- sided, and $P < 0.05$ was considered statistically significant.

No statistical test was used to determine appropriate sample size prior to data collection. Sample size was determined based on SNR of data collected, which directly impacts the confidence in fitting methods used, as shown in Supplementary Fig. 5. For studies with replicate measurements of the same imaging phantom, a sample size of three 10 min acquisitions was used. For studies involving discrete samples (human subject and animal tissue) each experimental group consisted of 5 individual samples, or 5 individual subjects. All data met a minimum SNR requirement of 20.

Standardized imaging phantoms were used to evaluate data reproducibility. Data was collected over the course of several weeks, in multiple locations (laboratory, athletic facility, hospital).

Data collection was performed randomly where possible. For studies involving imaging phantoms, care was taken to randomize the order of data collection by varying the phantom order and acquisition frequency to ensure replicates were not completed back-to-back. For studies involving animal tissue and human subjects, data was collected on the bases of subject recruitment and availability and animal tissue harvesting schedules. Blinding was not possible in this study. The study provides a characterization of device design and performance, for figures that demonstrate statistical significance between groups, the samples measured had very different physical characteristics and data could not be obtained blinded.

### Reporting summary

Further information on research design is available in the Nature Portfolio Reporting Summary linked to this article.

## Data availability

The figure data generated in this study have been deposited in the public figshare repository under accession code https://doi.org/10.6084/m9.figshare.24716313. Any additional requests for information can be directed to, and will be fulfilled by, the corresponding authors. Source data are provided with this paper.

## Code availability

Analysis code is available at https://doi.org/10.6084/m9.figshare.24716349.

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

## Acknowledgements
MIT IRB protocol 2002000099 was obtained to use our device on a cohort of healthy human subjects. We thank staff at the Koch Institute Swanson Biotechnology Center for support, specifically W. Huang at the Animal Imaging and Preclinical Testing Core; staff at the MIT DAPER Center for allowing us to collect human measurements in the athletic facility; Ashvin Bashyam and Chris Frangieh, Dr. Sagar Nigwekar, and Dr. Sahir Kalim for helpful discussions; and Natalie Ferris, Alicia D'Souza, and Artur Szałata for manuscript review. This work was supported by the National Institutes of Health grant no. R01EB031813 (M.J.C.). S.E.S. and A.S.Z. are NSF Graduate Research Fellows. M.S.R. acknowledges the generous support of the Kiyomi and Ed Baird MGH Research Scholar award.

## Author contributions
S.E.S. designed, constructed, and performed in-vitro characterization of the SSMR sensor. S.E.S., and A.S.Z. performed the in-vivo characterization. A.S.Z. performed the CuSO4 magnet comparison characterization. S.E.S. prepared the figures and wrote the manuscript. S.E.S., A.S.Z., W.-S.H, M.S.R. and M.J.C. edited the manuscript. M.S.R. and M.J.C. supervised the research and provided continual feedback.

## Competing interests
M.J.C. is an inventor on a patent (US10564237B2) and patent application (US20200383574A1) submitted by MIT that describes the design of the permanent magnet array. This interest is managed by MIT in accordance with their conflict of interest policies. Other authors have no competing interests to declare.
