## [Peer Review File · Nature Communications]

REVIEWER COMMENTS

Reviewer #1 (Remarks to the Author):

- What are the noteworthy results?

A low-field single-sided MR sensor was proposed to evaluate the skeletal muscle in vivo.

- Will the work be of significance to the field and related fields? How does it compare to the established literature? If the work is not original, please provide relevant references.

The work has a certain significance to the application field of single sided NMR. The proposed sensor is similar to other researchers', but uses more discretized magnets, which is somewhat innovative in the detection application of skeletal muscles. For the sensor itself, there was no special optimization, only ordinary parameter scanning methods were used (Line 296 The magnet size, orientation, location, spacing, etc were input parameters. These were swept to determine optimal magnet array design). For the analytical methods, a biexponential fit of the T2 decays was used. There are still some doubts about the analytical method of using the double exponential fitting of T2 decay, and further explanation is needed.

- Does the work support the conclusions and claims, or is additional evidence needed?

The additional evidence is needed to explain why the biexponential fit of the T2 decays is suitable for the detection of the skeletal muscle. The relationship between calculated T2 by biexponential fit of the CPMG decay and skeletal muscle evaluation is needed to provide. There is no theoretical description for this relationship, which is vital for the applicability of the designed probe.

- Are there any flaws in the data analysis, interpretation and conclusions? - Do these prohibit publication or require revision?

There are some errors or doubts that will be detailed in the following questions. These require revision.

- Is the methodology sound? Does the work meet the expected standards in your field?

The method still needs further explanation. This work can be further improved to meet the expected standards in our field.

- Is there enough detail provided in the methods for the work to be reproduced?

In the design method of sensor, as the parameter scanning method used is very common, these details are sufficient. From the method of biexponential fit of the T2 decays, the method is also commonly used and the details are sufficient. The processing of images and data statistics can provide some more details.

major points:

1. In this manuscript, the authors propose a low-field single-sided MR sensor, aiming for skeletal muscle in vivo. Although it was reported that similar MRI based systems and methods have been used for skeletal muscle evaluation, the low-field single-sided MR sensor designed in this manuscript presents a different way for skeletal muscle measurements. The manuscript is well written and the concept of sensor design is reasonable, but the method still needs further explanation. Some major concerns are needed to be addressed before it is acceptable.

2. In line 172, it states that " Data was collected from the phantoms at 8.48, 8.43, 8.38, and 8.29 MHz, corresponding to depths of approximately 2-, 5-, 8-, and 10-mm from the surface of the sensor...". In line 113, it states that " The mapped homogeneous region following fabrication has a field strength of 0.2 T and sits 8mm above the surface of the array with a natural descending gradient in the Z direction of 1 T/m...". These two paragraphs seem to contradict each other. Because the spin magnetic ratio of hydrogen protons is approximately 42.58 MHz/T, so the frequency changed 0.04258 MHz/mm. The phantoms at 8.48, 8.43, 8.38, and 8.29 MHz seem not corresponding to depths of approximately 2-, 5-, 8-, and 10-mm. Please check it.

In addition, the field strength of 0.2 T is corresponding to 8.516 MHz, then there will be an undeniable millimeter level error.

3. In Line 179 to 184, it states that "The signal from the layered phantom acquired at and below 8.38MHz is statistically the same as the muscle tissue phantom. This demonstrates we are only capturing the signal from the muscle portion of the layered phantom. The signal reflects amplitudes between the muscle and adipose phantoms at 8.43 MHz. The signal at this frequency contains contributions from both phantom types near the layer junction. The signal acquired at 8.48 MHz, however, statistically reflects the adipose phantom; verifying that we are fully below the phantom layer junction, we can achieve an accurate signal from muscle phantom above a 6-mm-thick layer of adipose phantom at 8.38 MHz using a biexponential fit of the decay..."

In fact, for unilateral magnets, due to the presence of natural gradients, it is relatively easy to choose a specific RF excitation frequency to determine which height the sample is excited to above the surface of the magnet. This has nothing to do with the biexponential fit of the decay method.

In addition, from the spectral analysis of Transverse relaxation time(T2), the composition of skeletal muscle or adipose layer containing hydrogen atom groups is also relatively complex, and they do not

correspond to a specific Transverse relaxation time(T_2). So the method of biexponential fit of the decay is also an approximate method. The T_2 spectra by ILT from the acquired CPMG echo curves could be used to as a further proof.

4. How to define the relationship between calculated T_2 by biexponential fit of the CPMG decay and skeletal muscle evaluation? There is no theoretical description for this relationship, which is vital for the applicability of the designed probe. Taking MRI for example, it directly gives a visual image showing the skeletal muscle status.

5. In Line 69, it states that "Previous clinical studies with SSMR sensors were limited by the penetration depth (< 6 mm) and signal sensitivity...". In fact, there are many SSMR sensors with penetration depth more than 6 mm and relatively high signal sensitivity. Previous clinical studies with SSMR sensors maybe limited by other reasons such as engineering issues.

6. In Line 154, it states that "Slice selection characterization was performed by adjusting the pulsed (B_1) frequency from 8.32 - 8.42 MHz in 0.1 MHz increments... " The 0.1 MHz increments should be 0.01 MHz increments from Figure 3 B.

7. For CPMG pulse, how about is the excitation bandwidth of the RF excitation pulse in kHz?

Reviewer #2 (Remarks to the Author):

This manuscript described a single-sided MR sensor for measuring calf muscle, with a main focus on the tailored magnet design. While each component of the system did not show any significant breakthroughs in terms of novelty, it did enable a particular yet limited in-vivo application. The verification experiments were reasonably designed and the conclusions were generally sound.

Specific comments:

1. FEA design. If the goal was to increase SNR, via trade-offs between sweet spot volume/ B_0 , and depth/weight etc., how exactly were the decisions made? Was there a quantifiable metric (e.g. SNR itself as a function of the outcome measures) that supported the statement and were any outcome measures play more important roles? These are important information currently missing.

2. The magnet design does not seem to deviate a lot from conventional single-sided Halbach arrays. What were the key differences that made the gradient smaller in the ROI when compared with "other

single-sided sensors”(Line 217-218)? This seems to be one highlight of the work and should be elaborated in greater details.

3. Line 389 – “A 16x16x 32 mm area in the center of the array containing the array sweet spot...- how large was the sweet spot volume relative to the size of the area? Was the RF bandwidth (both hardware limits and pulse length), hence the thickness, matched to the determined volume (presumably +/-0.5% of the center frequency)?

4. The targeted signal source is jointly selected by B0 and B1 fields. It would be great to show the 3D simulated excited slice (determined by center freq/BW) laying on top of a section of the calf (may be taken from a simplified model or even actual MR images), and see how much signal comes from the muscle but not the subcutaneous fat/tissues.

5. How was the size of the RF coil determined? It plays no lesser role in determining the ROI, particularly undesired signal from subcutaneous regions. It is important to show the 3D slice profile in 4 as it would inevitably cross the skin, just a matter of how far away from the coil sensitive areas and/or the flip angles.

6. How far was the coil placed above the magnet? Was ringdown a big issue?

7. Figure 3. Was the signal magnitude measured by the peak value? I think a more precise way would be calculate the area below the peak, as a flat but lower-amplitude spectrum could lead to more signal than a narrow but higher peak.

8. Figure 4. Why were the error bars much longer for 8.29/8.43MHz? Does that have any implications? Another thing is what was the diameter of the phantom and did that properly mimic the human calf geometry? Would it be possible to construct the phantom as two concentric rings (outer ring 6mm thickness) with the longitudinal direction covering the crossings of the slice profile and the phantom?

9. Figure 5B. Intuitively, the Human T2,1 should be closer to Rat Muscle T2,1 than Rat Fat T2,1. But that was not the case. Does that make sense?

10. How was the SNR affect the fitting? Line 429 – since the authors had all the data, a plot of fitting accuracy as a function of averaging numbers can be generated to provide some insights.

11. The diffusion effect was mentioned in the supplemental material but not in the main body. Did that provide poor differentiations compared with T2? Was the diffusion effect absorbed into T2 decay in the current simplified format, and if so, how large of the signal decay did it provide compared with T2?

The authors would like to express appreciation for the thorough review of our manuscript, “Single-sided magnetic resonance-based sensor for point-of-care evaluation of muscle”. The feedback has been invaluable in supplementing and refining the content and quality of the manuscript. The authors have carefully considered all comments and are grateful for the expertise and time dedicated to this process. In the following response, we address each of the points, showing how they have contributed to the improvement of the article.

Reviewer #1:

- What are the noteworthy results?

A low-field single-sided MR sensor was proposed to evaluate the skeletal muscle in vivo.

- Will the work be of significance to the field and related fields? How does it compare to the established literature? If the work is not original, please provide relevant references.

The work has a certain significance to the application field of single sided NMR. The proposed sensor is similar to other researchers', but uses more discretized magnets, which is somewhat innovative in the detection application of skeletal muscles. For the sensor itself, there was no special optimization, only ordinary parameter scanning methods were used (Line 296 The magnet size, orientation, location, spacing, etc were input parameters. These were swept to determine optimal magnet array design). For the analytical methods, a biexponential fit of the T2 decays was used. There are still some doubts about the analytical method of using the double exponential fitting of T2 decay, and further explanation is needed.

Response: Further justification for the use of biexponential fitting has been included in the Background section. Additional context and references to past and recent literature context and references has been included to provide support for the use of multi-compartment fitting models for quantitative T2 data.

Part of text adding into the Background section:

There is extensive evidence of skeletal muscle quantitative T2 relaxation being better represented by a bi-exponential model as compared to a mono-exponential model. There is open discussion as to the specific physiological context of the two exponential decays, with support for the two relaxation dynamics originating from either water and lipids, or from differing water compartments within a tissue¹²⁻¹⁴. Multi-compartment analysis has demonstrated higher specificity in differentiating muscle tissues with inflammatory pathologies, dystrophic pathologies, differing fat fractions, and differing water content, regardless of the origin of bi-exponential signal^{12,13}. We support the conclusions of previous work that the two relaxations represent intercellular (shorter component) and intracellular (longer component) water compartments within a singular tissue. While both muscle and subcutaneous tissues exhibit biexponential T2 decays, the shorter ‘intracellular’ time constant is largely conserved between tissue types, while the longer of the time constants differs between tissues and can be distinguished from one another¹⁴.

- Does the work support the conclusions and claims, or is additional evidence needed?

The additional evidence is needed to explain why the biexponential fit of the T2 decays is suitable for the detection of the skeletal muscle. The relationship between calculated T2 by biexponential fit of the CPMG decay and skeletal muscle evaluation is needed to provide. There is no theoretical description for this relationship, which is vital for the applicability of the designed probe.

Response: Same as above. Further information informing the decision to use a bi-exponential fit has been included in the manuscript.

- Are there any flaws in the data analysis, interpretation and conclusions? - Do these prohibit publication or require revision?

There are some errors or doubts that will be detailed in the following questions. These require revision.

- Is the methodology sound? Does the work meet the expected standards in your field?

The method still needs further explanation. This work can be further improved to meet the expected standards in our field.

Response: Several parts have been supplemented in the Methods section, including an additional subsection on FEA design and outcomes, and additional information pertaining to fitting and statistical analysis.

- Is there enough detail provided in the methods for the work to be reproduced?

In the design method of sensor, as the parameter scanning method used is very common, these details are sufficient. From the method of biexponential fit of the T2 decays, the method is also commonly used and the details are sufficient. The processing of images and data statistics can provide some more details.

Response: More details have been added in the Background and Methods sections on the data processing and analysis. A section on statistical analysis has been added.

major points:

1. In this manuscript, the authors propose a low-field single-sided MR sensor, aiming for skeletal muscle in vivo. Although it was reported that similar MRI based systems and methods have been used for skeletal muscle evaluation, the low-field single-sided MR sensor designed in this manuscript presents a different way for skeletal muscle measurements. The manuscript is well written and the concept of sensor design is reasonable, but the method still needs further explanation. Some major concerns are needed to be addressed before it is acceptable.

2. In line 172, it states that " Data was collected from the phantoms at 8.48, 8.43, 8.38, and 8.29 MHz, In line 113, it states that " The mapped homogeneous region following fabrication has a field strength of 0.2 T and sits 8mm above the surface of the array with a natural descending gradient in the Z direction of 1 T/m...". These corresponding to depths of approximately 2-, 5-, 8-, and 10-mm from the surface of the sensor...". two paragraphs seem to contradict each other. Because the spin magnetic ratio of hydrogen protons is approximately 42.58 MHz/T, so the frequency changed 0.04258 Mhz/mm. The phantoms at 8.48, 8.43, 8.38, and 8.29 MHz seem not corresponding to depths of approximately 2-, 5-, 8-, and 10-mm. Please check it.

In addition, the field strength of 0.2 T is corresponding to 8.516 MHz, then there will be an undeniable millimeter level error.

Response: This is an excellent point. The wording pertaining to the field strength has been clarified in the text. The maximum field strength of the array is 0.2T at the surface of the sensor. The homogeneous region has a slightly lower field strength (0.196T) that sits 8mm above the surface of the sensor. With this context clarified, the frequencies use for phantom experimentation correspond to the given depths.

3. In Line 179 to 184, it states that "The signal from the layered phantom acquired at and below 8.38MHz is statistically the same as the muscle tissue phantom. This demonstrates we are only capturing the signal from the muscle portion of the layered phantom. The signal reflects amplitudes between the muscle and adipose phantoms at 8.43 MHz. The signal at this frequency contains contributions from both phantom types near the layer junction. The signal acquired at 8.48 MHz, however, statistically reflects the adipose phantom; verifying that we are fully below the phantom layer junction, we can achieve an accurate signal from muscle phantom above a 6-mm-thick layer of adipose phantom at 8.38 MHz using a biexponential fit of the decay..."

In fact, for unilateral magnets, due to the presence of natural gradients, it is relatively easy to choose a specific RF excitation frequency to determine which height the sample is excited to above the surface of the magnet. This has nothing to do with the biexponential fit of the decay method.

In addition, from the spectral analysis of Transverse relaxation time(T2), the composition of skeletal muscle or adipose layer containing hydrogen atom groups is also relatively complex, and they do not correspond to a specific Transverse relaxation time(T2). So the method of biexponential fit of the decay is also an approximate method. The T2 spectra by ILT from the acquired CPMG echo curves could be used to as a further proof.

Response: We agree with the reviewer that the natural gradient of single sided magnets makes the selection of RF frequency for slice selection relatively easy. The use of bi-exponential fitting models does not contribute to the slice selection or determination of which tissue is being detected. Since skeletal muscle tissue is better modeled (detailed further in the Background in response to other comments) and since pathology detection in muscle is more sensitive with a bi-exponential fit, we use this fitting methods as our analysis for all data presented in this manuscript.

We agree that and ILT analysis of the curves could be used as further proof – currently our SNR is low for the use of ILT. In previous literature where ILT was implemented, 150 was the minimum SNR and even at an SNR of 500 it was 80% accurate detection of peaks (Ioannidis et al., 2020, Berman et al, 2013).

Ioannidis, G.S., Nikiforaki, K., Kalaitzakis, G. et al. Inverse Laplace transform and multiexponential fitting analysis of T2 relaxometry data: a phantom study with aqueous and fat containing samples. *Eur Radiol Exp* **4**, 28 (2020).

Berman, P., Levi, O., Parmet, Y., Saunders, M. and Wiesman, Z. (2013), Laplace inversion of low-resolution NMR relaxometry data using sparse representation methods. *Concepts Magn. Reson.*, 42: 72-88.

4. How to define the relationship between calculated T2 by biexponential fit of the CPMG decay and skeletal muscle evaluation? There is no theoretical description for this relationship, which is vital for the applicability of the designed probe Taking MRI for example, it directly gives a visual image showing the skeletal muscle status.

Response: Further details pertaining to the justification for applying multi-compartment models (bi-exponential fitting) to evaluation of skeletal muscle as well as further information on the applicability of the instrument has been included in the Background section. There is additionally now a caveat included of specific analysis techniques varying with the clinical application in question.

Additional text in the Background section includes:

Techniques including T2 relaxometry and T2-weighted diffusion can be performed on single-sided sensors to provide clinically-actionable information¹⁵⁻¹⁸. Uses include assessment of liver disease, inflammation, tumor characteristics, iron overload, and cartilage diseases, among others^{7,11,19}. Within skeletal muscle tissue specifically, relaxometry can provide insight into fluid status, progressive disease musculoskeletal disease monitoring (sarcopenia, muscular dystrophies, etc.), vascular kinetics and oxygenation tracking, among other applications^{10,13,20} No one clinical application is evaluated for the use of this tool. Outcome metrics, acquisition parameters, and analysis techniques for specific applications will vary based on clinical application.

5. In Line 69, it states that "Previous clinical studies with SSMR sensors were limited by the penetration depth (< 6 mm) and signal sensitivity...". In fact, there are many SSMR sensors with penetration depth more than 6 mm and relatively high signal sensitivity. Previous clinical studies with SSMR sensors maybe limited by other reasons such as engineering issues.

Response: This is an excellent point, the wording has been clarified and references included to point to further limitations, including engineering challenges, of SSMR sensors and their clinical applications.

6. In Line 154, it states that "Slice selection characterization was performed by adjusting the pulsed (B1) frequency from 8.32 - 8.42 MHz in 0.1 MHz increments... " The 0.1 MHz increments should be 0.01 MHz increments from Figure 3 B.

Response: Thank you for the correction. This mistake has been adjusted in the text.

7. For CPMG pulse, how about is the excitation bandwidth of the RF excitation pulse in kHz?

Response: The methods sections describing the pulse used now include the pulse bandwidth (43kHz).

Reviewer #2:

This manuscript described a single-sided MR sensor for measuring calf muscle, with a main focus on the tailored magnet design. While each component of the system did not show any significant breakthroughs in terms of novelty, it did enable a particular yet limited in-vivo application. The verification experiments were reasonably designed and the conclusions were generally sound.

Specific comments:

1. FEA design. If the goal was to increase SNR, via trade-offs between sweet spot volume/B0, and depth/weight etc., how exactly were the decisions made? Was there a quantifiable metric (e.g. SNR itself as a function of the outcome measures) that supported the statement and were any outcome measures play more important roles? These are important information currently missing.

Response: This is an excellent point – an additional section has been added to the Methods section further detailing design process, outcome metrics, and key decisions influencing the final array design

Additional text includes:

A two-step approach was used to determine array configuration. Initially 2-D models were used to approximate the net magnetic profile of several basic magnet array geometries including unilateral Halbach, semi-cylindrical Halbach, U shaped, and L shaped

For these geometries, an optimization score was calculated based on the B0 strength and gradient: $Optimization\ Score = \frac{B_0^7}{RMSE * G_0}$

³⁵. The Optimization score and depth of homogeneous region was used to select a semi-cylindrical design as the basis for further design optimization. A 3-D model of the geometry was then used to evaluate several parameters including magnet size, shape,

orientation, spacing, inset depths, and iron yoke position and determine the effects of the parameters on the magnetic field profile. The position of the iron yoke on the inner sides of the raised Halbach elements had the strongest effect on increasing the area, depth, and strength of the field. The raised Halbach elements on the ends of the array and the magnetic orientation, particularly of the outer slices of magnets, similarly had a strong effect on the final profile. Features including the size of the magnet cubes, central slice, and inset depths of the central slice had opposing effects on the net field strength, area, and depth.

2. The magnet design does not seem to deviate a lot from conventional single-sided Halbach arrays. What were the key differences that made the gradient smaller in the ROI when compared with “other single-sided sensors”(Line 217-218)? This seems to be one highlight of the work and should be elaborated in greater details.

Response: More details have been included in the Discussion section specifying the specific design aspects that contributed to the reduced gradient.

Additional text includes:

We maximize the signal acquisition region by minimizing the gradient. This is accomplished by raising the edge Halbach elements of the magnet array into more of a semi-cylindrical geometry and positioning iron yokes on the inner surfaces of these edge elements, which collectively reduce the gradient in the homogeneous region while maintaining its depth from the array surface.

3. Line 389 – “A 16x16x 32 mm area in the center of the array containing the array sweet spot...- how large was the sweet spot volume relative to the size of the area? Was the RF bandwidth (both hardware limits and pulse length), hence the thickness, matched to the determined volume (presumably +/-0.5% of the center frequency)?

Response: The Methods section detailing mapping now includes the sweet spot volume and slice thickness. Further, the excitation bandwidth has been added to the Methods section detailing the pulses used. The bandwidth does match the +/-0.5% of the homogeneous region.

4. The targeted signal source is jointly selected by B0 and B1 fields. It would be great to show the 3D simulated excited slice (determined by center freq/BW) laying on top of a section of the calf (may be taken from a simplified model or even actual MR images), and see how much signal comes from the muscle but not the subcutaneous fat/tissues.

Response: An additional figure has been included that highlights subject positioning and the excited slice on an MRI. Additionally, the potential limitation of variable body composition on the tissue contained in the homogeneous region has been added to the discussion.

5. How was the size of the RF coil determined? It plays no lesser role in determining the ROI, particularly undesired signal from subcutaneous regions. It is important to show the 3D slice profile in 4 as it would inevitably cross the skin, just a matter of how far away from the coil sensitive areas and/or the flip angles.

Response: The 3D slice profile is now included as supplemental figure 6. The size of the surface RF coil was selected to balance the limitations of larger diameter surface coils, with higher penetration into the sample but with weaker fields and larger inhomogeneities in the plane of the coil, with smaller coils that can produce higher fields with less coil-plane inhomogeneities. We selected a 16mm diameter coil, which allows for repeatable excitation and

slice selection, and operates towards the upper limits of our RF amplifier. Future work is planned in developing novel coil design for use with single side sensors.

6. How far was the coil placed above the magnet? Was ringdown a big issue?

Response: Further detail on the coil placement and points excluded due to ringdown has been included in the body of the Results section and the methods section.

7. Figure 3. Was the signal magnitude measured by the peak value? I think a more precise way would be to calculate the area below the peak, as a flat but lower-amplitude spectrum could lead to more signal than a narrow but higher peak.

Response: The calculation of the peak signal magnitude is now included in the Methods section. The peak is calculated by averaging points 3-13 of the acquired decay. The initial 3 points are discarded.

8. Figure 4. Why were the error bars much longer for 8.29/8.43MHz? Does that have any implications? Another thing is what was the diameter of the phantom and did that properly mimic the human calf geometry? Would it be possible to construct the phantom as two concentric rings (outer ring 6mm thickness) with the longitudinal direction covering the crossings of the slice profile and the phantom?

Response: Further details about the phantom geometry is included in the results section. The size of the phantom does exceed the size of the coil ensuring that the entire sensitive region is 'filled' with phantom. There is ongoing exploration of phantom construction and geometry that will be the subject of a separate manuscript. The error bars are longer for signal acquired at 8.29MHz due to the distance from the sensor surface (>10mm) and higher B0 gradient at that depth resulting in higher variability in the fit results despite a comparable SNR. All data for this figure was collected in a clinical environment rather than a lab environment to reflect a more realistic performance, there are a few trials (including one at 8.43MHz) with a significantly lower SNR as a result of environmental activity in the vicinity. In that case, the SNR did affect the fit of the data. This is further addressed for point 10 with an additional figure.

9. Figure 5B. Intuitively, the Human T_{2,1} should be closer to Rat Muscle T_{2,1} than Rat Fat T_{2,1}. But that was not the case. Does that make sense?

Response: We support based on prior literature that the T_{2,1} component reflects the intracellular compartment of tissue and that among different tissue types, this value would be conserved. There is prior work of bi-exponential fitting on MRI data demonstrating a range of T₂ for both subcutaneous fat and skeletal muscle having values from 20-70ms with tails of the T_{2,1} distribution ranging from 5-90 ms. The T_{2,1} data shown in figure 5B does fall within these expected ranges for intracellular compartments, but due to tight confidence intervals makes each set statistically different from each other. Further exploration of expected intra-individual variations within these ranges is needed to assert more.

10. How does the SNR affect the fitting? Line 429 – since the authors had all the data, a plot of fitting accuracy as a function of averaging numbers can be generated to provide some insights.

Response: An additional supplementary figure with the suggested analysis has been included. The figure demonstrates the effect of SNR on bi-exponential fit results.

Supplementary figure 5. Effect of SNR on fit results. Bi-exponential fit results for muscle phantom a) T2,1 b) T2,2 c) Amp1, and d) Amp2 with increased SNR.

11. The diffusion effect was mentioned in the supplemental material but not in the main body. Did that provide poor differentiations compared with T2? Was the diffusion effect absorbed into T2 decay in the current simplified format, and if so, how large of the signal decay did it provide compared with T2?

Response: Thank you for catching this – the methods section on diffusion has been removed. Diffusion applications of the sensor will be addressed in a future manuscript.

REVIEWERS' COMMENTS

Reviewer #1 (Remarks to the Author):

Authors have addressed the main concerns in the submission, and the related problems in the original manuscript have solved. I recommend to accept the paper in its current form.

Reviewer #2 (Remarks to the Author):

I hope the authors can elaborate more on my previous point #4.

* I understand that from Fig S6, most of the signals come from the red mark. But the excited slice extends beyond it (and will cross the subcutaneous tissues). If the assumption is the signal is negligible outside the marked sweet spot due to the stronger gradient, would it be possible to quantify that percentage to confirm?

* B1 effect (independent of gradient) - Anything outside of the sweet spot but are closer to the coil could have higher B1(Rx). For example, if the excited slice and skin/fat crossing is closer to the coil, the signal might not be negligible even if the gradient is higher. This was also part of the reason I suggested a complete 3D excited slice containing the skin crossings.

* Fig S3 - label fonts are too small to read.

Other than that, I am satisfied with the authors' response.

Response to reviewer comments:

The authors would like to thank the reviewer for their continued support in making the manuscript as accurate as possible. The final comments have been individually addressed in the manuscript, figures, and below.

* I understand that from Fig S6, most of the signals come from the red mark. But the excited slice extends beyond it (and will cross the subQ tissues). If the assumption is the signal is negligible outside the marked sweet spot due to the stronger gradient, would it be possible to quantify that percentage to confirm?

A: This is true, although based on experimental data we assume a negligible signal outside the sweet spot, it is unlikely there is zero influence from signal from outside the sweet spot. A percentage calculation would be difficult, but we can quantify that an additional one millimeter of tissue outside the sweet spot is likely to have a stronger influence on signal. The exact tissue included in this region is person-dependent based on differences in anatomy. This has been added to the discussion.

* B1 effect (independent of gradient) - Anything outside of the sweet spot but are closer to the coil could have higher B1(Rx). For example, if the excited slice and skin/fat crossing is closer to the coil, the signal might not be negligible even if the gradient is higher. This was also part of the reason I suggested a complete 3D excited slice containing the skin crossings.

A: Similar as above, we cannot assume zero influence on signal from outside the sweet spot due to B1 inhomogeneity, especially given the current pulse shape and sequence used for signal acquisition. Figure S6 and the discussion have been updated to reflect this. This is an area of active exploration in the lab and will be the topic of a future publication.

* Fig S3 - label fonts are too small to read.

A: Fonts have been increased for better readability